molecular biology/cellular biology/biophysics

haematopoiesis, histone demethylase, transcription factors, protein–protein interaction

**Authors for correspondence:**
Catherine Porcher
e-mail: catherine.porcher@imm.ox.ac.uk
Erika J. Mancini
e-mail: erika.mancini@sussex.ac.uk

# The histone H3K4 demethylase JARID1A directly interacts with haematopoietic transcription factor GATA1 in erythroid cells through its second PHD domain

Dimple Karia[1], Robert C. G. Gilbert[1],
Antonio J. Biasutto[1,2], Catherine Porcher[3]
and Erika J. Mancini[1,4]

[1]Division of Structural Biology, Wellcome Trust Centre for Human Genetics, University of Oxford, Roosevelt Drive, Oxford OX3 7BN, UK
[2]Department of Biochemistry, University of Oxford, 3 S Parks Road, Oxford OX1 3QU, UK
[3]MRC Molecular Haematology Unit, Weatherall Institute of Molecular Medicine, University of Oxford, Oxford OX3 9DS, UK
[4]School of Life Sciences, University of Sussex, Falmer, Brighton BN1 9RH, UK

DK, 0000-0002-3776-1650; RCGG, 0000-0001-9336-5604;
EJM, 0000-0001-9591-7898

Chromatin remodelling and transcription factors play important roles in lineage commitment and development through control of gene expression. Activation of selected lineage-specific genes and repression of alternative lineage-affiliated genes result in tightly regulated cell differentiation transcriptional programmes. However, the complex functional and physical interplay between transcription factors and chromatin-modifying enzymes remains elusive. Recent evidence has implicated histone demethylases in normal haematopoietic differentiation as well as in malignant haematopoiesis. Here, we report an interaction between H3K4 demethylase JARID1A and the haematopoietic-specific master transcription proteins SCL and GATA1 in red blood cells. Specifically, we observe a direct physical contact between GATA1 and the second PHD domain of JARID1A. This interaction has potential implications for normal and malignant haematopoiesis.

# 1. Introduction

Epigenetic modifications, such as histone methylation, play critical roles in gene expression and regulate many cellular processes [1]. Histone lysine methylation is involved in transcriptional regulation through the combined action of histone methylases and demethylases. Specifically, methylated histone H3 lysine 4 (H3K4) marks contribute to active transcription, as emphasized by the observation that the three methylation states (H3K4me1, H3K4me2 and H3K4me3) mark actively transcribed genes. However, while a clear correlation exists between H3K4 methylation and active transcription, less is known about how specific methyltransferases or demethylases are recruited to specific gene loci.

JARID1A (also known as KDM5A or RBP2) belongs to the KDM5 family of histone demethylases. It is an approximately 200 kDa, ubiquitously expressed nuclear protein [2] that specifically demethylates tri- and dimethylated H3K4 [3]. Given the role of H3K4 methylation in activating transcription, it is thought that JARID1A acts as a transcriptional repressor through its chromatin demethylase activity. For example, studies have shown that JARID1A represses transcription in ES cells by binding to the promoter region of the homeobox genes *Hoxa1*, *a5* and *a7* [3]. Increasing evidence from primary tumours and model systems supports a role for the KDM5 family as oncogenic drivers [4], but their contribution to the mechanisms leading to malignant transformation remain poorly investigated. Indeed, although JARID1A was initially isolated based on its association with Retinoblastoma (Rb) tumour suppressor protein decades ago [5], its mechanistic role in oncogenesis is still unclear.

Functionally, JARID1A is involved in normal and malignant haematopoiesis. Peripheral blood analysis of JARID1A knock-out mice showed neutrophilia, and analysis of their haematopoietic stem cell and myeloid compartments revealed a significant decrease in the rate of apoptosis as well as enhanced survival and cell cycling [6]. Moreover, JARID1A has been identified as a fusion partner of Nucleoporin-98 (NUP98, a common fusion partner found in many leukaemias) in a subset of acute myeloid leukaemia (AML) patients in which chromosomal translocation of *NUP98* resulted in fusion of its amino terminus with the C-terminus PHD motif (PHD3) of JARID1A [7] (figure 1). This fusion protein arrests haematopoietic differentiation in murine models and induces AML by abrogating the H3K4me3 binding potential of PHD3 finger [8]. NUP98/JARID1A fusion is found in about 11% of acute megakaryoblastic leukaemia (AMKL) patients, who, interestingly, also present overexpression of *HOXA* and *HOXB* gene clusters [9].

As well as mediating fusion with NUP98, PHD3 is thought to mediate interactions of JARID1A to the LIM domain only 2 protein in murine erythroleukaemia (MEL) cells [5]. LMO2 is a member of the so-called LIM-only class of transcriptional co-regulators, which act as protein–protein adaptors and mediate interactions through their LIM domains [10,11]. LMO2 is a core component of a haematopoietic master transcriptional regulator complex known as the pentameric complex, which also includes SCL/TAL1 and its heterodimerization partner E47, LDB1 and GATA1 [12] (figure 1). The crystal structure of the DNA-bound SCL/E47 basic helix–loop–helix (bHLH) heterodimer in complex with LMO2 fused to the LDB1 LIM interaction domain (LID) revealed complex synergies between components of the complex, their cofactors and DNA targets [13].

This complex is a key driver of gene expression programmes in both embryonic and adult haematopoiesis [14]. Emphasizing its role in erythropoiesis, homozygous mutations of SCL/TAL1, LMO2 or GATA1 genes all result in embryonic lethality between days E9–E11.5 due to complete lack of formation of red blood cells [15–17]. In T cells, aberrant expression of SCL and LMO2 as a result of chromosomal translocation among other oncogenic processes is involved in the development of acute lymphoblastic leukaemia (T-ALL) [18,19].

In the context of the pentameric complex, SCL serves both as a transcriptional activator and repressor in a context-dependent manner [14]. This dual activity results from recruitment of both coactivators, like acetyltransferases P300 and PCAF [20,21], and corepressors such as ETO2, mSin3A, HDAC1/2 and LSD1 [22–25]. Specifically, phosphorylation-mediated dynamic recruitment of LSD1, another histone demethylase involved in removing monomethylated or dimethylated H3K4 (also called KDM1A), modulates SCL-directed transcription of target genes that are essential for haematopoietic progenitor differentiation [25]. Knockdown (KD) of LSD1 in haematopoietic progenitors impairs differentiation of multiple haematopoietic lineages, indicating that LSD1 has a critical role in haematopoiesis [24]. While much is known about the interactions of LSD1 with SCL, the role of other histone demethylases in this context has not been investigated.

Given the documented interactions of JARID1A with LMO2, JARID1A's implication in AML and the fact that many protein partners of SCL, such as mSin3A and pRB [22,26], are also interaction partners of JARID1A [27,28], we set out to investigate a potential physical interaction between JARID1A and the pentameric protein complex to gain insights into JARID1A's function in haematopoiesis. Our results

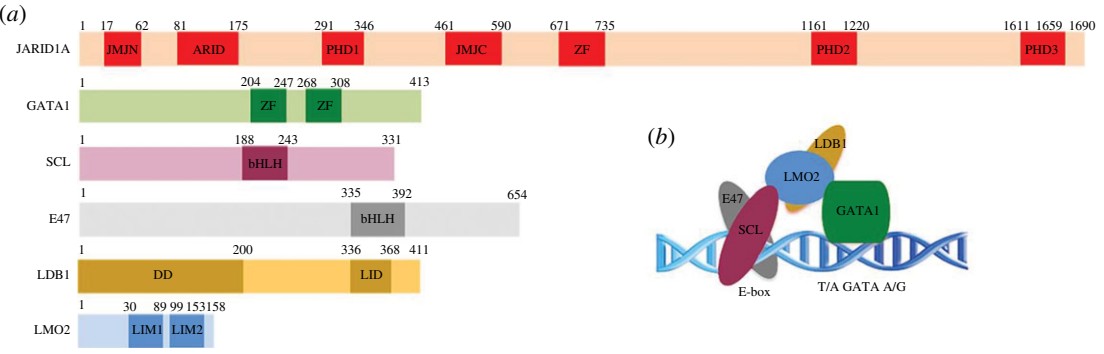

**Figure 1.** Schematic of JARID1A and SCL–GATA1 complex. (*a*) Schematic of the domains found in JARID1A, GATA1, SCL, E47, LDB1 and LMO2. JmjN, jumonji N; Arid, AT-rich interacting domain; PHD, plant homeo domain; JmjC, jumonji C; ZF, zinc finger; bHLH, basic helix–loop–helix; LID, LIM interaction domain; LIM, Lin11/Isl1/Mec3 domain. The numbering for all proteins refers to the human sequence. (*b*) Model of the SCL–GATA1 complex bound to DNA.

indicate that in erythroid cells, JARID1A associates with specific members of pentameric complex, including SCL and GATA1, and that it directly interacts with GATA1 through its second PHD domain.

# 2. Material and methods

## 2.1. Cell culture

MEL (clone 585) cells were maintained in Dulbecco's modified Eagle medium-high glucose (Gibco) supplemented with 10% FCS, 100 U ml$^{-1}$ penicillin, 100 µg ml$^{-1}$ streptomycin and 2 mM L-glutamine. MEL cells differentiation was induced by addition of 2% DMSO followed by incubation at 37°C for 4 days.

## 2.2. Nuclear extract preparations

For nuclear extract preparation, MEL cells were harvested by centrifugation at 1000*g* and washed once in 1× phosphate-buffered saline. Cell pellets were resuspended in chilled resuspension buffer (10 mM HEPES, pH 7.9, 1.5 mM MgCl$_2$, 10 mM KCl, Complete EDTA free protease inhibitor (Roche), 1 mM dithiotreitol (DTT)). Cell suspension was incubated on ice for 10 min, mixed occasionally and vortexed for 10 s. The nuclei were pelleted by centrifugation for 1 min at 16 300*g* and resuspended in 20 mM HEPES, pH 7.9, 1.5 mM MgCl$_2$, 420 mM NaCl, 25% glycerol, Complete EDTA free protease inhibitor (Roche), 1 mM DTT and incubated for 1 h at 4°C on a rotary shaker. Nuclear extracts were obtained by centrifugation at 16 300*g* for 5 min and supernatants aliquoted and stored at −80°C.

## 2.3. Size-exclusion chromatography

The 2.5 mg crude nuclear extracts isolated from MEL cells were subjected to fractionation on a Superose 6H/R column (Amersham Biosciences) equilibrated in 20 mM HEPES, pH 7.9, 200 mM NaCl, 1 mM DTT, 0.2 mM PMSF, 10% glycerol. At a flow rate of 0.5 ml min$^{-1}$, 0.5 ml fractions were collected, precipitated with 72% trichloroacetic acid and subjected to western blotting.

## 2.4. Western blotting

Western blotting analysis was performed using NuPAGE precast gels (3–8% Tris-acetate for JARID1A and 4–12% Bis–tris for SCL, GATA1, LMO2 and LDB1; Life Technologies) according to the manufacturer's instructions. Primary antibodies used were: JARID1A (ab70892; Abcam), SCL (sc-12984; Santa Cruz), LDB1 (sc-11198; Santa Cruz), LMO2 (MCA2744GA, ABD serotec) and GATA1 (sc-1234; Santa Cruz). Secondary antibodies used were horseradish peroxidase conjugated anti-goat/mouse/rabbit immunoglobulin (Santa Cruz). Protein detection was carried out using ECL prime reagent kit (Amersham).

## 2.5. Coimmunoprecipitations and immunoblot analysis

One milligram of nuclear extracts was diluted in dilution buffer (50 mM Tris, pH 7.5, 0.3% NP-40, protease inhibitor, 1 mM DTT) to attain a final concentration of 150 mM NaCl. Nuclear extracts were pre-cleared for 3 h at 4°C by incubation with normal IgG (goat sc-2028 and rabbit sc-2027; Santa Cruz) and protein G Dynabeads (Life Technologies), blocked with 0.2 mg ml$^{-1}$ BSA (Pierce) and 0.4 mg ml$^{-1}$ sonicated salmon sperm DNA (Life Technologies). Immunoprecipitations were performed overnight at 4°C by incubation of the pre-cleared nuclear extracts with primary antibodies (for JARID1A, ab 70892; Abcam) and blocked protein G Dynabeads in dilution buffer (50 mM Tris, pH 7.5, 0.3% NP-40, protease inhibitor, 1 mM DTT). Beads were washed five times with 4× bed volume of wash buffer (50 mM Tris, pH 7.5, 150 mM NaCl, 0.3% NP-40, protease inhibitor, 1 mM DTT) and bound material was eluted in 1× Laemmli buffer by boiling at 95°C for 10 min. Fractions were analysed by immunoblotting as described above.

## 2.6. Cloning, expression and purification of JARID1A and GATA1 for analytical ultracentrifugation

The three PHD domains of JARID1A (referred to as PHD1 (282–360), PHD2 (1156–1222) and PHD3 (1608–1659), respectively, in this paper) were amplified using JARID1A mouse cDNA (GeneArt) as template and primers 5′-aagttctgtttcagggcccgCGGAAGGGCACCCTGAGCG-3′/5′-atggtctagaaagctttaCTCGCGCACGGCCTGCTC-3′ (for PHD1), 5′-aagttctgtttcagggcccgGAACGGATCGAGGAAGTGAAGTTCTGC-3′/5′-atggtctagaaagctttaTCTGGGCCGTCTGCTCCG-3′ (for PHD2) and 5′-aagttctgtttcagggcccgGCCGTGTGTGCCGCCCAG-3′/5′-atggtctagaaagctttaGGCGCAGTTGATGCAGATGTAGTCC3′ (for PHD3) and subsequently cloned using ligation independent In-fusion system (Clontech) into pOPINJ or pOPINF vectors [29], encoding, respectively, either an N-terminal hexahistidine tag followed by a glutathione S-transferase (GST) tag or only an N-terminal hexahistidine (His) tag. The human SCL(180–253)/E47 (535–613) bHLH was cloned, expressed and purified as described [13]. The human GATA1 construct (encoding N- and C-terminus fingers preceded by an N-terminal hexahistidine tag and referred to as GATA1NFCF (201–313) in this paper) was a generous gift from Dr Olga Platonova.

The GST-Tag PHD1, PHD2 and PHD3 constructs were transformed into *Escherichia coli* BL21 cells for protein expression. Cells were grown at 37°C in Luria Broth (LB) until absorbance at 600 nm reached 0.6. This was followed by induction of protein expression by addition of 1 mM isopropyl β-D-1-thiogalactopyranoside (IPTG) and growth at 20°C for 20 h. Cell lysis was carried out in lysis buffer (50 mM HEPES, pH 8, 400 mM NaCl, 0.5% Tween 20, 34 µl of sigma protease inhibitor, 12.5 units of DNaseI from Sigma) and the clarified lysate was mixed with glutathione beads. This was followed by overnight incubation at 4°C on a rotary shaker. Next day, beads were washed with the wash buffer (50 mM HEPES, pH 8, 400 mM NaCl). The over-expressed protein was eluted using the elution buffer (50 mM HEPES, pH 8, 200 mM NaCl and 15 mM reduced glutathione). The eluted protein fractions were concentrated using a concentrator (Millipore) with a molecular weight cut-off (MWCO) appropriately chosen on the basis of the molecular weight of the protein. The concentrated proteins were further purified by size-exclusion chromatography, by loading on a HiLoad 16/60 Superdex 75 prep grade column (GE Healthcare) in 50 mM HEPES, pH 8, 200 mM NaCl, 1 mM DTT.

The PHD2 and GATA1NFCF constructs were transformed into *E. coli* Rosetta (DE3) pLysS cells for protein expression. Cells were grown at 37°C in LB until absorbance at 600 nm reached 0.6. This was followed by induction of protein expression by addition of 1 mM IPTG and growth at 20°C for 16–20 h. Cells transformed with plasmid encoding GATA1NFCF were grown in LB autoinduction media [30] overnight at 37°C. Cells were harvested by centrifugation and resuspended in 50 mM HEPES, pH 8, 400 mM NaCl, 10 mM Imidazole, 0.5% (v/v) Tween 20, protease inhibitor and DNase I (for GATA1NFCF, lysis buffer was supplemented with 1 mM β-mercapto ethanol and 5 mM MgCl$_2$). Cells were disrupted by sonication on ice and cell lysate clarified by centrifugation at 48 384$g$. The clarified cell lysate was incubated with Talon affinity resin (Clontech) equilibrated with lysis buffer for an hour at 4°C. The beads were subsequently washed with 10 bed volumes of 50 mM HEPES, pH 8, 400 mM NaCl, 10 mM Imidazole (for GATA1NFCF: 50 mM HEPES, pH 8, 400 mM NaCl, 30 mM Imidazole, 1 mM β-mercapto ethanol and 5 mM MgCl$_2$) and the protein was eluted in 50 mM HEPES, pH 8, 200 mM NaCl, 500 mM Imidazole (for GATA1NFCF, elution buffer was supplemented with 1 mM β-mercapto ethanol and 5 mM MgCl$_2$). The eluant was concentrated using protein concentrator (Millipore) of appropriate MWCO (3 kDa for PHD2 and 10 kDa for GATA1NFCF). Both the proteins were further purified by size-exclusion chromatography using pre-equilibrated Superdex 75 (16/60)

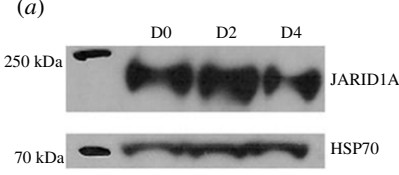

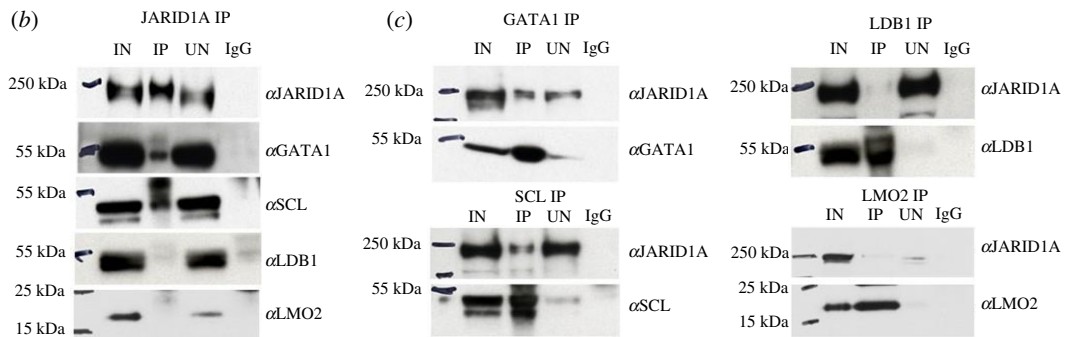

**Figure 2.** Endogenous expression of JARID1A and coimmunoprecipitation of JARID1A, GATA1, SCL, LDB1 and LMO2. (*a*) JARID1A endogenous expression was checked at day 0 (D0), day 2 (D2) and day 4 (D4) of MEL cell differentiation by western blotting. Heat shock protein 70 (HSP 70) was used as loading control. (*b*) Nuclear extracts from MEL cells were immunoprecipitated with JARID1A antibodies and the precipitated proteins were analysed by western blotting using antibodies shown. (*c*) Reverse coimmunoprecipitations were carried out on nuclear extracts from MEL cells with antibodies indicated and the precipitated proteins were analysed by western blotting.

(GE Healthcare) in 50 mM HEPES, pH 8, 200 mM NaCl, 1 mM DTT. The purified proteins were assessed by SDS–PAGE and concentrated to be used for analytical ultracentrifugation (AUC).

## 2.7. Analytical ultracentrifugation

Analytical ultracentrifugation sedimentation velocity experiments were performed using a Beckman Optima XL-I analytical ultracentrifuge operating in velocity mode at 20°C. All protein samples, at a molar concentration of 0.04 mM, were loaded onto 3 mm double sector-shaped centrepieces and spun at 29 024$g$ (An60Ti rotor, Beckman Coulter Inc., CA, USA) with 150 sample distribution scans being taken with an interval of 6 min using 280 nm absorbance optics. The 10–120th scans were analysed using SedFit software [31] using continuous $c(s)$ model to calculate the sedimentation coefficients of the sedimenting species at a resolution of 100 and also using $c(s,f/f_0)$ method at a resolution of 10. The frictional coefficient is a unitless entity and is the ratio between the experimental frictional coefficient and the minimum frictional coefficient if the molecules in the sample were compact spheres ($f/f_0$). Analysed data were then plotted using Pro Fit software (QuantumSoft; Uetikon am See, Switzerland) and fitted with a Gaussian function to determine peak centres, which represent the sedimentation coefficients of the species. Sedimentation coefficient ($s$) was plotted on the $X$-axis, continuous sedimentation coefficient distribution ($c(s)$) on the $Y1$-axis, molecular weight on the $Y2$-axis and $c(s,f/f_0)$ was plotted over a third dimension which was represented by a heat plot. All the samples were in 50 mM HEPES, pH 8, 200 mM NaCl and 1 mM DTT.

# 3. Results

## 3.1. JARID1A interacts with the pentameric complex in erythroid cells

We first interrogated the possible association between JARID1A and the pentameric complex in the cell line MEL, a well-established *in vitro* model of erythroid differentiation. Expression of endogenous JARID1A was detected in both undifferentiated MEL cells (D0, pro-erythroblast stage) and MEL cells undergoing terminal erythroid maturation (D2 and D4) (figure 2*a*). As SCL and protein partners are known to engage repressive activities in erythroid progenitors to prevent precocious lineage differentiation [14,23], we explored the relationship between JARID1A and the SCL complex in undifferentiated MEL cells. Nuclear extracts

were derived from MEL cells and coimmunoprecipitation experiments were performed with antibodies detecting endogenous proteins. Immunoprecipitation (IP) with antibodies against JARID1A followed by immunoblotting (IB) with antibodies against SCL, LMO2, LDB1 or GATA1 demonstrated that JARID1A coimmunoprecipitated with SCL and GATA1 but not LMO2 or LDB1 (figure 2b). Reciprocally, IP with antibodies against the components of the pentameric complex and IB with antibodies against JARID1A also revealed that SCL and GATA1, but not LDB1 or LMO2 coimmunoprecipitated with JARID1A (figure 2c). This is in contrast with previously published findings where JARID1A was found to be co-immunoprecipitated with *in vitro* translated radio-labelled LMO2 in MEL cells nuclear extracts [5]. It is possible that weak interactions exist between LMO2 and JARID1A that can be detected by a more sensitive radioactive but not by chemiluminescent assay. Protein fractionation carried out from MEL cell nuclear extracts by fast protein liquid chromatography with a high salt extraction and size-exclusion approach followed by western blotting shows JARID1A eluting in a number of fractions with a range of apparent molecular mass much greater than that of the monomeric protein (192 kDa), suggesting that JARID1A may be part of a multiprotein complex (electronic supplementary material, figure S1). Although the elution pattern of JARID1A overlaps in many fractions with that of GATA1 and SCL, bands can also be seen for LMO2 and LDB1 suggesting that the resolution of the technique does not allow an accurate delineation of the composition of such JARID1A containing complexes.

## 3.2. JARID1A interacts directly with GATA1

In order to determine the molecular basis for the interaction of JARID1A with members of the pentameric complex, we first carefully selected the domains of each of these proteins likely to be involved in protein–protein interactions based on previous reports. We then analysed interactions between the selected domains by AUC.

JARID1A, like other KDM5 family members, possesses an atypical insertion of a DNA-binding ARID (AT-rich) domain and a histone-binding PHD (PHD1) domain into the catalytic Jumonji domain, which separates it into two fragments (JmjN and JmjC) (figure 1). It also contains two further C-terminal PHD domains (PHD2 and PHD3). PHD domains are structures known to mediate protein–protein interactions [32]. In particular, JARID1A PHD3 has been studied in the context of its fusion with nucleoporin NUP98, where it has been shown to lose its binding to the H3K4 tri-methyl marks following fusion [7,8]. The PHD1 domain preferentially recognizes unmethylated H3K4 histone tails, the product of JARID1A-mediated H3K4me3 demethylation [33]. No function for the PHD2 domain has been reported to date, and its ability to bind histone tails has not been probed. Separately, a number of PHD domains have been identified that can also bind non-histone protein targets. For example, the PHD3 finger of MLL1 can associate with H3K4me2/3 and with the RNA recognition motif (RRM) of CyP33 using distinct binding pockets [34]. Therefore, JARID1A PHD1–3 domains were selected for analysis.

Previous functional and structural data indicate that SCL/E47 bHLH and GATA1 zinc-finger domains mediate protein–protein interactions with ETO2/p300 [13] and Fli-1/FOG-1, respectively [35,36], suggesting these domains as candidates for interaction with JARID1A. Additionally, in the case of SCL, structure/function studies have shown that the bHLH domain alone can functionally rescue blood development in an *Scl*-null background [37], thus strengthening its functional relevance. We thus focused our analysis on the SCL/E47 bHLH domain and GATA1 zinc fingers.

AUC was used to probe direct interactions between GST-fused JARID1A PHD domains and either HIS-tagged SCL(180–253)/E47(535–613) bHLH domains (SCL/E47 bHLH) or HIS-tagged GATA1 zinc-finger domains (GATA1NFCF) (201–313). We first determined the apparent sedimentation coefficients of GST PHD1 (282–360) and SCL/E47 bHLH separately, which showed sedimentation coefficients of $3.8 \pm 0.09$ and $1.9 \pm 0.04$ S (Svedbergs), respectively (figure 3a). An equimolar mixture of both proteins resulted in sedimentation coefficient peaks at $1.9 \pm 0.08$ and $4.3 \pm 0.17$ S (figure 3a). The first peak corresponds to unbound SCL/E47 bHLH, and while the second peak shows an increase in sedimentation coefficient. The model independent $c(s,ff_0)$ and molecular weight analysis shows three species at $1.8 \pm 0.27$, $3.9 \pm 0.2$ and $4.5 \pm 0.2$ S representing unbound SCL/E47 bHLH, GST PHD1 and complex, respectively (figure 3d). The presence of three peaks suggests a mixture of different states and a dynamic interaction between SCL/E47 and GST PHD1. GST PHD2 (1156–1222) and SCL/E47bHLH individually showed sedimentation coefficients values of $3.8 \pm 0.20$ and $1.8 \pm 0.10$ S, respectively (figure 3b). When mixed in equimolar amounts, two well-defined peaks at $1.8 \pm 0.10$ and $3.8 \pm 0.19$ S can be observed, which overlap with the coefficients of the individual protein (figure 3b). This together with the model independent $c(s, ff_0)$ and molecular weight analysis (figure 3e) suggests no complex formation. Apparent sedimentation coefficient analysis of GST PHD3 (1608–1659) and SCL/E47 bHLH showed peaks at $3.6 \pm 0.07$ and $1.9 \pm$

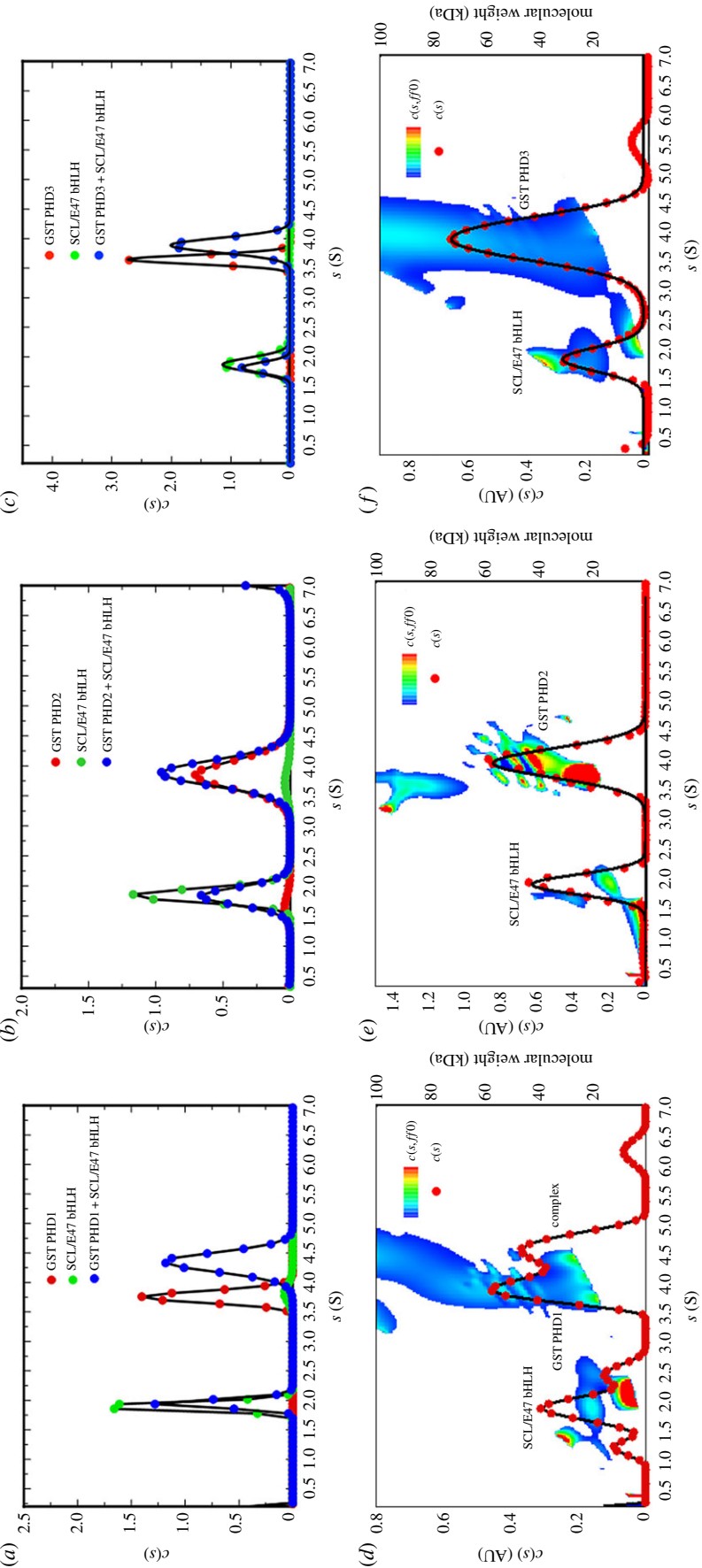

**Figure 3.** Biophysical studies of interaction between JARID1A and SCL/E37 by AUC. AUC apparent sedimentation coefficient distribution profiles superimposed of (*a*) SCL/E47 bHLH (green), GST PHD1 (red) and SCL/E47 bHLH+ GST PHD1 (blue). (*b*) SCL/E47 bHLH (green), GST PHD2 (red) and SCL/E47 bHLH+ GST PHD2 (blue). (*c*) SCL/E47 bHLH (green), GST PHD3 (red) and SCL/E47 bHLH+GST PHD3 (blue). $c(s, ff_0)$ and molecular weight analysis for (*d*) GST PHD1 and SCL/E47bHLH, (*e*) GST PHD2 and SCL/E47bHLH and (*f*) GST PHD3 and SCL/E47 bHLH. The *Y*2-axis shows the estimated molecular weights and the temperature plot represents the distribution of the apparent frictional coefficients in the sample, which is plotted over a third dimension (see key for contour plots).

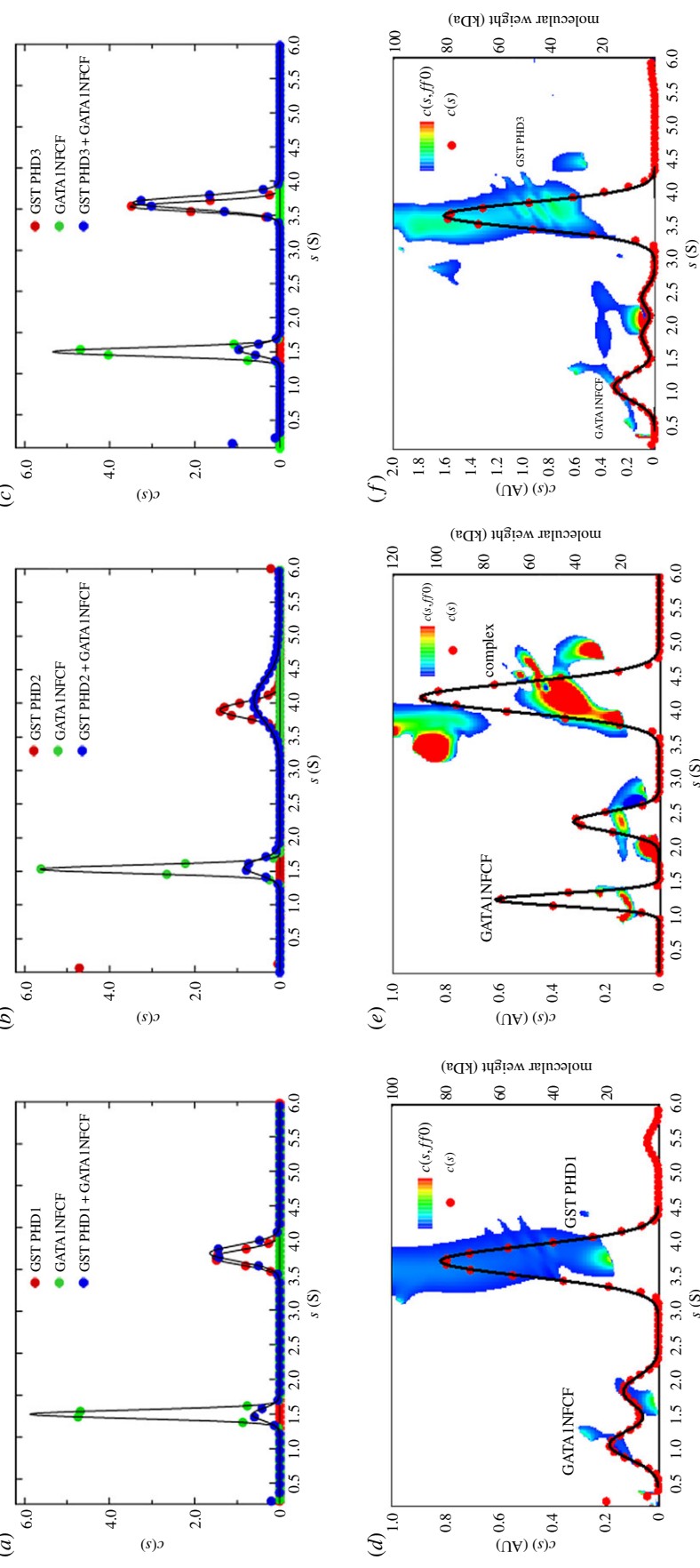

**Figure 4.** Biophysical studies of interaction between JARID1A and GATA1 by AUC. AUC apparent sedimentation coefficient distribution profiles superimposed of (*a*) GATA1NFCF (green), GST PHD1 (red) and GATA1NFCF+GST PHD1 (blue), (*b*) GATA1NFCF (green), GST PHD2 (red) and GATA1NFCF+GST PHD2 (blue) and (*c*) GATA1NFCF (green), GST PHD3 (red) and GATA1NFCF +GST PHD3 (blue). *c*(*s*,*ff0*) and molecular weight analysis for (*d*) GST PHD1 and GATA1NFCF, (*e*) GST PHD2 and GATA1NFCF and (*f*) GST PHD3 and GATA1NFCF. The *Y2*-axis shows the estimated molecular weights and the temperature plot represents the distribution of the apparent frictional coefficients in the sample, which is plotted over a third dimension (see key for contour plots).

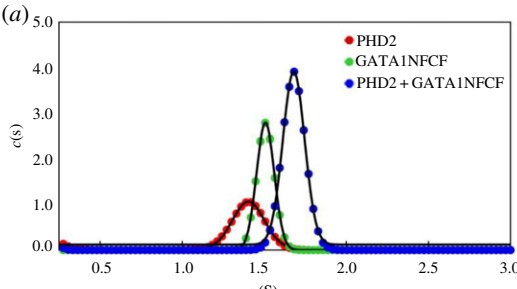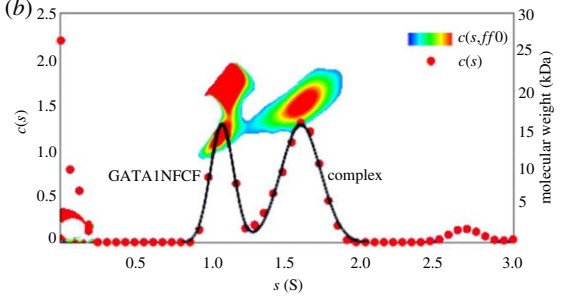

**Figure 5.** Biophysical studies of interaction between PHD2 domain of JARID1A and GATA1 by AUC. (*a*) AUC apparent sedimentation coefficient distribution profiles of HIS-PHD2 (red), GATA1NFCF (green) and HIS-PHD2+GATA1NFCF (blue) superimposed. (*b*) $c(s,ff_0)$ and molecular weight analysis for HIS-PHD2+GATA1NFCF. The $Y2$-axis shows the estimated molecular weights and the temperature plot represents the distribution of the apparent frictional coefficients in the sample which is plotted over a third dimension (see key for contour plots).

0.06 S, respectively (figure 3*c*). When mixed in equimolar amounts, two well-defined peaks at $1.9 \pm 0.06$ and $3.8 \pm 0.10$ S can again be observed (figure 3*c*). Together with the model independent $c(s,ff_0)$ and molecular weight analysis (figure 3*f*), they suggest no complex formation.

Next, we investigated interactions between the zinc-finger domains of GATA1 and the PHD domains of JARID1A. We first determined the apparent sedimentation coefficients of GST PHD1 and GATA1NFCF separately, which showed values of $3.8 \pm 0.10$ and $1.5 \pm 0.06$ S (figure 4*a*). An equimolar mixture of the two proteins yielded two peaks with values of again $1.5 \pm 0.08$ and $3.8 \pm 0.10$ S overlapping with the individual proteins' peaks, indicating no complex formation (figure 4*a,d*). GST PHD2 and GATA1NFCF separately displayed peaks at $3.9 \pm 0.10$ and $1.5 \pm 0.06$ S, respectively (figure 4*b*). When mixed in equimolar amounts, two well-defined peaks appeared at $1.5 \pm 0.10$ and $4.0 \pm 0.30$ S (figure 4*b*). The first peak probably originates from unbound GATA1NFCF, while the second broader peak extends further than the sedimentation coefficient peak of GST PHD2 alone. This result, together with the model independent $c(s,ff_0)$ and molecular weight analysis (figure 4*e*), strongly suggests an interaction between GST PHD2 and GATA1NFCF. Finally, GST PHD3 and GATA1NFCF separately displayed peaks at $3.6 \pm 0.07$ and $1.5 \pm 0.06$ S, respectively (figure 4*c*). An equimolar mixture of the two proteins yielded two peaks overlapping with the individual protein peaks (figure 4*c*) which, together with the model independent $c(s,ff_0)$ and molecular weight analysis (figure 4*f*), suggests no complex formation between GST PHD3 and GATA1NFCF.

To exclude any interference of the GST affinity tag, we further probed the interactions of a His-tag PHD2 domain of JARID1A. First, we used $^{15}N$ HSQC NMR to probe the correct folding of the HIS-tag PHD2 construct. Well-dispersed, individual peaks can be distinguished in the 2D NMR spectrum (electronic supplementary material, figure S2) suggesting a folded protein. We then used AUC to study the apparent sedimentation coefficients of His-tag PHD2 and GATA1NFCF. HIS-tagged PHD2 and GATA1NFCF separately showed sedimentation coefficients of $1.4 \pm 0.10$ and $1.5 \pm 0.05$ S, respectively (figure 5*a*). An equimolar mixture of both proteins showed a sedimentation coefficient peak at $1.7 \pm 0.06$ S, a shift that is highly suggestive of complex formation (figure 5*a*). The model independent $c(s,ff_0)$ and molecular weight analysis of absorbance data obtained from the equimolar mixture of PHD2 and GATA1NFCF resulted in two peaks (figure 5*b*). The first peak has a molecular weight of 15 kDa probably originating from unbound GATA1NFCF, while the second peak has a molecular weight of 20 kDa, probably deriving from the complex.

In conclusion, AUC experiments have revealed that the second PHD (PHD2) domain of JARID1A interacts directly with the zinc-finger domains of GATA1. Also, the data show that there is a possible direct interaction between the first PHD domain of JARID1A and the bHLH domains of SCL/E47, although this is likely to be highly dynamic in nature. As we have probed the interactions of the PHD (1–3) domains of JARID1A and, respectively, the bHLH domains of SCL/E47 and zinc-finger domains of GATA1, these experiments do not rule out interactions involving other parts of the proteins.

## 4. Discussion

Histone demethylase JARID1A is a recognized key factor for haematopoietic cell differentiation and proliferation and its disruption in the form of a fusion protein is prevalent in AML patients. Fusion

proteins containing chromatin remodelling proteins often target histone-modifying activities to inappropriate genomic targets. An outstanding biological and mechanistic question is how JARID1A is normally targeted to the correct genomic loci.

In this study, we report the interaction between JARID1A and members of the haematopoietic-specific pentameric protein complex (SCL and GATA1) in erythroid cells. When analysing putative interactions between known protein–protein interaction domains of JARID1A, SCL/E47 and GATA1, we show that this interaction could be directly mediated by GATA1 zinc-finger domains and the second PHD domain (PHD2) of JARID1A. Interestingly, the loss of both normal GATA1 and JARID1A function is linked with AMKL [9,38,39], suggesting a possible functional link between loss of interaction and leukemogenesis.

In conclusion, our data suggest that, in haematopoietic cells, JARID1A could be targeted to its genomic loci through interactions with a key component of the essential pentameric complex and thus participate in the control of expression of haematopoietic-specific genes and blood lineage differentiation. Further studies are required to elucidate whether JARID1A PHD2 domain is able to bind concomitantly histone tails and GATA1 and to further understand the functional relevance of JARID1A interaction with the SCL/GATA1 pentameric complex in the context of normal and aberrant haematopoiesis.

Data accessibility. All data supporting this article are available within the article or have been uploaded as part of the electronic supplementary material.

Authors' contributions. D.K. designed the experiments, acquired the data, analysed the data, interpreted the data and drafted the paper; R.C.G.G. acquired the data, analysed the data and interpreted the data; D.K. and A.J.B. performed and analysed the NMR experiments; C.P. conceived and designed the experiments, analysed the data, interpreted the data and revised the paper; E.J.M. conceived and designed the experiments, analysed the data, interpreted the data and drafted the paper.

Competing interests. We have no competing interests.

Funding. This work was supported by Bloodwise (Gordon Pillar studentship to E.J.M. no. 09039) and the Medical Research Council (grant no. MC_UU_12009/9 to C.P.). The Division of Structural Biology is a part of the Wellcome Centre for Human Genetics, Wellcome Trust Core grant no. 090532/Z/09/Z.

Acknowledgements. We are grateful to Dr Olga Platonova, Dr Tao Ni and Dr Sarah Hoosdally for help and advice during the experimental work. We gratefully acknowledge the support of Dr David Staunton with the use of the Oxford Molecular Biophysics Suite within the University's Department of Biochemistry. We would like to thank Prof. Christina Redfield for help with the processing of the NMR data.

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
