## [Reviewer comments · Royal Society Open Science]

Review History

RSOS-191048.R0 (Original submission)

Review form: Reviewer 1

Is the manuscript scientifically sound in its present form?

Yes

Are the interpretations and conclusions justified by the results?

No

Is the language acceptable?

Yes

Do you have any ethical concerns with this paper?

No

Have you any concerns about statistical analyses in this paper?

No

Recommendation?

Accept with minor revision (please list in comments)

Comments to the Author(s)

The manuscript by Karia et al. describes the identification of an interaction between the H3K4 histone demethylase JARID1A and the key transcription factors SCL and GATA1 in murine erythroid cells. The authors employ analytical ultracentrifugation to show a direct contact between the PHD1 domain of JARID1A and the zinc fingers of GATA1 and, possibly between PHD1 and the SCL/E47 bHLH domain of SCL. The experimental work is well designed and presented and the data are, overall, clear and convincing. The work described is of potential significance due to the implication of both JARID1A and GATA1 in Acute Megakaryoblastic Leukemia (AMKL). However, the specific points discussed below should be considered before the manuscript is accepted for publication:

1. The gel filtration profiles presented in Figure 2B do not add much to the weight of the manuscript, particularly in light of the co-IP data in Figure 3 which show lack of interaction between JARID1A and LDB1 and, possibly, with LMO2. As such, Fig. 2B should be moved to Supplementary data and Figure 2A merged with Figure 3.
2. The model shown in Figure 5C is not supported by the co-IP experiments of Figure 3 which showed a lack of interactions between JARID1A and LDB1 and LMO2. Also the statement in the abstract that reporting "an interaction between H3K4 demethylase JARID1A and..... SCL, E47, LMO2, LDB1 and GATA1..." again is not supported by the co-IP data of Figure 3. The authors would need to address these discrepancies.
3. The panels in Figure 4 have been mislabeled in relation to the main text and the legend, i.e. panels A-C should become D-F and vice versa.

Review form: Reviewer 2

Is the manuscript scientifically sound in its present form?

No

Are the interpretations and conclusions justified by the results?

No

Is the language acceptable?

Yes

Do you have any ethical concerns with this paper?

No

Have you any concerns about statistical analyses in this paper?

No

Recommendation?

Major revision is needed (please make suggestions in comments)

Comments to the Author(s)

In this manuscript, the authors describe an interaction between the hematopoietic transcription factor GATA1 and the lysine demethylase JARID1A.

First, the authors show a gel filtration trace from MEL cell nuclear extract, where they have western blotted the fractions for all of the proteins of interest. They state, referring to the portion of JARID1A that doesn't elute in the void, that:

"Significantly, the elution pattern of JARID1A largely overlapped with that of GATA1, SCL, LDB1 and LMO2"

To my eye, however, JARID1A really doesn't overlap significantly with GATA1. The GATA1 lies in a single peak centred at ~14-14.5 ml, whereas the JARID1A is centred at 13 ml and is undetectable at 14.5 ml. In fact, evidence for the pentameric complex is not strong at all and I don't agree with the way that the authors have labelled peaks underneath the western blots – they indicate a peak centred at 15 ml, but really there is only a very minor LDB1 peak centred at that elution time. LDB1 and LMO2 are predominantly eluting in a complex centred at 12-12.5 ml (that must contain proteins that are not being blotted for here). Overall, this figure doesn't really seem to support the authors' thesis.

Next, the authors show coIP experiments, IP-ing each of the relevant proteins in turn. Interactions between JARID1A and GATA1 and SCL are indicated (though they may be direct or indirect at this stage), but no significant interaction with LDB1 or LMO2 is seen (though the authors take the view that LMO2 *is* interactions – seems pretty marginal though).

In the last part of the manuscript, the authors use analytical ultracentrifugation to try to determine whether a domain from JARID1A (they choose the three PHD domains for their experiments) interacts directly with the structured domains from either GATA1 (the double ZF domain) or SCL (the bHLH domain), the two proteins that gave positive results in the coIPs (although it is odd that the authors claim that LMO2 is an interacting protein from the coIPs but then do not test this protein by AUC – specifically by sedimentation velocity).

First, the authors show AUC data for each of the three PHD domains with the GATA-NFCF and with SCL-bHLH [see below for comments about whether the SCL is a GST fusion protein...]. They see a clear interaction between PHD1 and SCL, but dismiss this as "weak transient or unspecific" because they say that they see three peaks at 1.8 (SCL), 3.9 (GST-PHD1) and 4.5 (complex) S.

Now, there is a bit of ambiguity in Fig 4 because it seems that the panels somehow in the wrong place, but I can't see any panels that have a peak distribution like that for the mixture. The one that is labelled as PHD1 + SCL shows two peaks for the mixture – I can't see any free GST-1PHD1 peak in the panel (blue trace in panel D). As far as I can make out, this suggests that there was an excess of SCL in the mixture that was prepared, but that all of the PHD1 has been converted to a complex. And – even if there *were* three peaks, I don't think it is appropriate to dismiss the interaction as unspecific, just because the affinity might be weak – many biologically relevant interactions have affinities of even as low as mM (eg electron transport proteins). In the area of gene regulation, many acetyllysine-bromodomain interactions are 10-100 uM.

As an aside, these data highlight another issue with the approach taken here. There is no attempt to quantify the interactions (sed. velocity isn't really suited to affinity measurements, as far as I know) – so one can't even really say whether a situation that gave rise to three peaks would correspond to a 'weak' interaction because we don't get any sense of what 'weak' would mean in this context.

Continuing on in Figure 4, PHD2 clearly doesn't form a complex with SCL. The data with PHD3 are interpreted as not showing an interaction. The sed. coeff. is clearly a bit larger for one of the peaks in the mixture (in contrast to the PHD2 data, which are pretty clear), but I don't have a good enough sense of the uncertainties associated with sed. velocity data to know whether this difference is significant or not.

They then repeat these experiments with GATA1. No interaction is seen with PHD1 or PHD3, but a change in the sedimentation profile *is* observed with PHD2, indicating an interaction. Again, to my eye, panel B looks very much like panel D, and so it is hard not to draw the conclusion that PHD1 is interacting with SCL. It can't be *that* unspecific, because PHD1 doesn't interact with GATA1.

Finally, in Figure 5, the authors show sed. velocity data for a mixture of GATA1-NFCF and PHD2, which nicely show the formation of a complex.

Overall, the concerns that I have raised above and below mean that I do not believe that this manuscript is suitable for publication as it stands. As well as the various mixups that I mention, my main concerns are (a) that the gel filtration data really don't support the authors' claims and

that (b) the authors do not show any data that speak to the correctly folded nature of the domains that they work with in the AUC experiments. If these issues can be addressed, I would consider the manuscript suitable for publication in the journal (although, *really*, I'd prefer to see some other biophysical characterization data, given that the authors have been able to make a pair of isolated domains that behave well enough to collect AUC data...).

Additional comments

Note that there is an inconsistency in the text regarding the state of the PHD domains. Figure 4 (and the Results section) indicates that the PHD domains are present as GST fusions in the AUC, whereas the M+M text describes them as being cloned into as His-tag vector. No M+M is given for the generation and purification of GST-fusion proteins. If the PHD domains *are* GST fusions (and their sedimentation coefficients seem to indicate that this is the case), this raises some concerns in the AUC, given the micromolar dimerization constant of GST.

I think we really need to see some sort of indication that the purified domains are correctly folded – such as a 1D ¹H NMR spectrum, for example – or, in the case of GATA1 and SCL domains, an EMSA. It would also have been nice, given that the authors have used the AUC here, to see sedimentation equilibrium data – at least for the domains that are proposed to interact (alone and together).

I don't really think at this stage that the data are sufficiently convincing to make the case that the authors wish to make.

Minor points

Summary sentence 1 seems almost a tautology: 'transcriptional regulation controls gene expression'.

Line 70 – the authors need to fix “need Wadman et al. Embo 1997”

M+M – spaces needed before units.

M+NM – the authors need to describe how their nuclear extracts were prepared.

The caption for Figure 1 needs to state whether the numbering shown is for the human proteins or another species.

Line 470 – correct “Western blotting Western blotting analysis” (add full stop)

Figure 3 – the authors should show sufficient of the western blots to include the nearest molecular weight marker – and indicate this marker.

Line 243 – what do the authors mean by “post fusion”?

Line 249 – the authors need to add a reference.

The panels of Figure 4 are incorrectly arranged, compared to the figure legend and the article text.

The authors do not indicate the molar concentrations of the proteins used in AUC shown in Figure 4 (only that shown in Fig 5). Were they the same in every experiment in Figure 4 and 5?

They state only in the M+M that two of the proteins were used at 1 mg/ml (which isn't really the best units to use, given that it is the molar concentrations that are relevant).

Decision letter (RSOS-191048.R0)

18-Sep-2019

Dear Professor Mancini,

The editors assigned to your paper ("H3K4 demethylase JARID1A directly interacts with haematopoietic transcription factor GATA1 in erythroid cells through its second PHD domain") have now received comments from reviewers. We would like you to revise your paper in accordance with the referee and Associate Editor suggestions which can be found below (not

including confidential reports to the Editor). Please note this decision does not guarantee eventual acceptance.

Please submit a copy of your revised paper before 11-Oct-2019. Please note that the revision deadline will expire at 00.00am on this date. If we do not hear from you within this time then it will be assumed that the paper has been withdrawn. In exceptional circumstances, extensions may be possible if agreed with the Editorial Office in advance. We do not allow multiple rounds of revision so we urge you to make every effort to fully address all of the comments at this stage. If deemed necessary by the Editors, your manuscript will be sent back to one or more of the original reviewers for assessment. If the original reviewers are not available, we may invite new reviewers.

- Data accessibility

<http://datadryad.org/submit?journalID=RSOS&manu=RSOS-191048>

- Competing interests

- Authors' contributions

All submissions, other than those with a single author, must include an Authors' Contributions section which individually lists the specific contribution of each author. The list of Authors should meet all of the following criteria; 1) substantial contributions to conception and design, or

acquisition of data, or analysis and interpretation of data; 2) drafting the article or revising it critically for important intellectual content; and 3) final approval of the version to be published.

- Acknowledgements

- Funding statement

Kind regards,
Lianne Parkhouse
Royal Society Open Science
openscience@royalsociety.org

on behalf of Professor Alan Warren (Associate Editor) and Catrin Pritchard (Subject Editor)
openscience@royalsociety.org

Reviewers' Comments to Author:

Reviewer: 1

The manuscript by Karia et al. describes the identification of an interaction between the H3K4 histone demethylase JARID1A and the key transcription factors SCL and GATA1 in murine erythroid cells. The authors employ analytical ultracentrifugation to show a direct contact between the PHD1 domain of JARID1A and the zinc fingers of GATA1 and, possibly between PHD1 and the SCL/E47 bHLH domain of SCL. The experimental work is well designed and presented and the data are, overall, clear and convincing. The work described is of potential significance due to the implication of both JARID1A and GATA1 in Acute Megakaryoblastic Leukemia (AMKL). However, the specific points discussed below should be considered before the manuscript is accepted for publication:

1. The gel filtration profiles presented in Figure 2B do not add much to the weight of the manuscript, particularly in light of the co-IP data in Figure 3 which show lack of interaction between JARID1A and LDB1 and, possibly, with LMO2. As such, Fig. 2B should be moved to Supplementary data and Figure 2A merged with Figure 3.
2. The model shown in Figure 5C is not supported by the co-IP experiments of Figure 3 which showed a lack of interactions between JARID1A and LDB1 and LMO2. Also the statement in the abstract that reporting "an interaction between H3K4 demethylase JARID1A and..... SCL, E47,

LMO2, LDB1 and GATA1..." again is not supported by the co-IP data of Figure 3. The authors would need to address these discrepancies.

3. The panels in Figure 4 have been mislabeled in relation to the main text and the legend, i.e. panels A-C should become D-F and vice versa.

Reviewer: 2

In this manuscript, the authors describe an interaction between the hematopoietic transcription factor GATA1 and the lysine demethylase JARID1A.

First, the authors show a gel filtration trace from MEL cell nuclear extract, where they have western blotted the fractions for all of the proteins of interest. They state, referring to the portion of JARID1A that doesn't elute in the void, that:

"Significantly, the elution pattern of JARID1A largely overlapped with that of GATA1, SCL, LDB1 and LMO2"

To my eye, however, JARID1A really doesn't overlap significantly with GATA1. The GATA1 lies in a single peak centred at ~14-14.5 ml, whereas the JARID1A is centred at 13 ml and is undetectable at 14.5 ml. In fact, evidence for the pentameric complex is not strong at all and I don't agree with the way that the authors have labelled peaks underneath the western blots - they indicate a peak centred at 15 ml, but really there is only a very minor LDB1 peak centred at that elution time. LDB1 and LMO2 are predominantly eluting in a complex centred at 12-12.5 ml (that must contain proteins that are not being blotted for here). Overall, this figure doesn't really seem to support the authors' thesis.

Next, the authors show coIP experiments, IP-ing each of the relevant proteins in turn. Interactions between JARID1A and GATA1 and SCL are indicated (though they may be direct or indirect at this stage), but no significant interaction with LDB1 or LMO2 is seen (though the authors take the view that LMO2 *is* interactions - seems pretty marginal though).

In the last part of the manuscript, the authors use analytical ultracentrifugation to try to determine whether a domain from JARID1A (they choose the three PHD domains for their experiments) interacts directly with the structured domains from either GATA1 (the double ZF domain) or SCL (the bHLH domain), the two proteins that gave positive results in the coIPs (although it is odd that the authors claim that LMO2 is an interacting protein from the coIPs but then do not test this protein by AUC - specifically by sedimentation velocity).

First, the authors show AUC data for each of the three PHD domains with the GATA-NFCF and with SCL-bHLH [see below for comments about whether the SCL is a GST fusion protein...]. They see a clear interaction between PHD1 and SCL, but dismiss this as "weak transient or unspecific" because they say that they see three peaks at 1.8 (SCL), 3.9 (GST-PHD1) and 4.5 (complex) S.

Now, there is a bit of ambiguity in Fig 4 because it seems that the panels somehow in the wrong place, but I can't see any panels that have a peak distribution like that for the mixture. The one that is labelled as PHD1 + SCL shows two peaks for the mixture - I can't see any free GST-1PHD1 peak in the panel (blue trace in panel D). As far as I can make out, this suggests that there was an excess of SCL in the mixture that was prepared, but that all of the PHD1 has been converted to a complex. And - even if there *were* three peaks, I don't think it is appropriate to dismiss the interaction as unspecific, just because the affinity might be weak - many biologically relevant interactions have affinities of even as low as mM (eg electron transport proteins). In the area of gene regulation, many acetyllysine-bromodomain interactions are 10-100 uM.

As an aside, these data highlight another issue with the approach taken here. There is no attempt to quantify the interactions (sed. velocity isn't really suited to affinity measurements, as far as I know) - so one can't even really say whether a situation that gave rise to three peaks would correspond to a 'weak' interaction because we don't get any sense of what 'weak' would mean in this context.

Continuing on in Figure 4, PHD2 clearly doesn't form a complex with SCL. The data with PHD3 are interpreted as not showing an interaction. The sed. coeff. is clearly a bit larger for one of the peaks in the mixture (in contrast to the PHD2 data, which are pretty clear), but I don't have a

good enough sense of the uncertainties associated with sed. velocity data to know whether this difference is significant or not.

They then repeat these experiments with GATA1. No interaction is seen with PHD1 or PHD3, but a change in the sedimentation profile **is** observed with PHD2, indicating an interaction. Again, to my eye, panel B looks very much like panel D, and so it is hard not to draw the conclusion that PHD1 is interacting with SCL. It can't be **that** unspecific, because PHD1 doesn't interact with GATA1.

Finally, in Figure 5, the authors show sed. velocity data for a mixture of GATA1-NFCF and PHD2, which nicely show the formation of a complex.

Overall, the concerns that I have raised above and below mean that I do not believe that this manuscript is suitable for publication as it stands. As well as the various mixups that I mention, my main concerns are (a) that the gel filtration data really don't support the authors' claims and that (b) the authors do not show any data that speak to the correctly folded nature of the domains that they work with in the AUC experiments. If these issues can be addressed, I would consider the manuscript suitable for publication in the journal (although, **really**, I'd prefer to see some other biophysical characterization data, given that the authors have been able to make a pair of isolated domains that behave well enough to collect AUC data...).

Additional comments

Note that there is an inconsistency in the text regarding the state of the PHD domains. Figure 4 (and the Results section) indicates that the PHD domains are present as GST fusions in the AUC, whereas the M+M text describes them as being cloned into as His-tag vector. No M+M is given for the generation and purification of GST-fusion proteins. If the PHD domains **are** GST fusions (and their sedimentation coefficients seem to indicate that this is the case), this raises some concerns in the AUC, given the micromolar dimerization constant of GST.

I think we really need to see some sort of indication that the purified domains are correctly folded – such as a 1D 1H NMR spectrum, for example – or, in the case of GATA1 and SCL domains, an EMSA. It would also have been nice, given that the authors have used the AUC here, to see sedimentation equilibrium data – at least for the domains that are proposed to interact (alone and together).

I don't really think at this stage that the data are sufficiently convincing to make the case that the authors wish to make.

Minor points

Summary sentence 1 seems almost a tautology: 'transcriptional regulation controls gene expression'.

Line 70 – the authors need to fix "need Wadman et al. Embo 1997"

M+M – spaces needed before units.

M+NM – the authors need to describe how their nuclear extracts were prepared.

The caption for Figure 1 needs to state whether the numbering shown is for the human proteins or another species.

Line 470 – correct "Western blotting Western blotting analysis" (add full stop)

Figure 3 – the authors should show sufficient of the western blots to include the nearest molecular weight marker – and indicate this marker.

Line 243 – what do the authors mean by "post fusion"?

Line 249 – the authors need to add a reference.

The panels of Figure 4 are incorrectly arranged, compared to the figure legend and the article text.

The authors do not indicate the molar concentrations of the proteins used in AUC shown in Figure 4 (only that shown in Fig 5). Were they the same in every experiment in Figure 4 and 5? They state only in the M+M that two of the proteins were used at 1 mg/ml (which isn't really the best units to use, given that it is the molar concentrations that are relevant).

Author's Response to Decision Letter for (RSOS-191048.R0)

See Appendix A.

Decision letter (RSOS-191048.R1)

09-Dec-2019

Dear Professor Mancini,

It is a pleasure to accept your manuscript entitled "H3K4 demethylase JARID1A directly interacts with haematopoietic transcription factor GATA1 in erythroid cells through its second PHD domain" in its current form for publication in Royal Society Open Science. The comments of the reviewer(s) who reviewed your manuscript are included at the foot of this letter.

Kind regards,
Anita Kristiansen
Editorial Coordinator
Royal Society Open Science
openscience@royalsociety.org

on behalf of Professor Alan Warren (Associate Editor) and Catrin Pritchard (Subject Editor)
openscience@royalsociety.org

Associate Editor Comments to Author (Professor Alan Warren):
The authors have satisfactorily addressed the comments of the reviewers

Appendix A

We are very grateful to the Reviewers for their constructive comments, which we feel have improved our manuscript. Below we address the comments point-by-point.

Response to Reviewers' comments:

Reviewer 2

The manuscript by Karia et al. describes the identification of an interaction between the H3K4 histone demethylase JARID1A and the key transcription factors SCL and GATA1 in murine erythroid cells. The authors employ analytical ultracentrifugation to show a direct contact between the PHD1 domain of JARID1A and the zinc fingers of GATA1 and, possibly between PHD1 and the SCL/E47 bHLH domain of SCL. The experimental work is well designed and presented and the data are, overall, clear and convincing. The work described is of potential significance due to the implication of both JARID1A and GATA1 in Acute Megakaryoblastic Leukemia (AMKL). However, the specific points discussed below should be considered before the manuscript is accepted for publication:

Q1.1. The gel filtration profiles presented in Figure 2B do not add much to the weight of the manuscript, particularly in light of the co-IP data in Figure 3 which show lack of interaction between JARID1A and LDB1 and, possibly, with LMO2. As such, Fig. 2B should be moved to Supplementary data and Figure 2A merged with Figure 3.

A1.1: As suggested, we have moved Fig. 2B to Supplementary data (Supplementary Fig.1) and we have merged Fig.2A with Fig.3.

Q1.2. The model shown in Figure 5C is not supported by the co-IP experiments of Figure 3 which showed a lack of interactions between JARID1A and LDB1 and LMO2. Also the statement in the abstract that reporting "an interaction between H3K4 demethylase JARID1A and..... SCL, E47, LMO2, LDB1 and GATA1..." again is not supported by the co-IP data of Figure 3. The authors would need to address these discrepancies.

A1.2: To address the discrepancies we have removed the model from Figure 5 and we have modified the statement in the abstract from "interaction between H3K4 demethylase JARID1A and SCL, E47, LMO2, LDB1 and GATA1..." to "interaction between H3K4 demethylase JARID1A and the haematopoietic-specific master transcription proteins SCL and GATA1 in red blood cells".

Q1.3. The panels in Figure 4 have been mislabeled in relation to the main text and the legend, i.e. panels A-C should become D-F and vice versa.

A1.3: We apologize for the mislabelling. We have now rectified this. Also, as explained in **A2.4** below we have now split this Figure 4 into two new figures (Fig 3 and Fig4) incorporating data from Supplementary Fig.1 and 2.

Reviewer 2

Q2.1 In this manuscript, the authors describe an interaction between the hematopoietic transcription factor GATA1 and the lysine demethylase JARID1A. First, the authors show a gel filtration trace from MEL cell nuclear extract, where they have western blotted the fractions for all of the proteins of interest. They state, referring to the portion of JARID1A that doesn't elute in the void, that: "Significantly, the elution pattern of JARID1A largely overlapped with that of GATA1, SCL, LDB1 and LMO2" To my eye, however, JARID1A really doesn't overlap significantly with GATA1. The GATA1 lies in a single peak centred at ~14-14.5 ml, whereas the JARID1A is centred at 13 ml and is undetectable at 14.5 ml. In fact, evidence for the pentameric complex is not strong at all and I don't agree with the way that the authors have labelled peaks underneath the western blots – they indicate a peak centred at 15 ml, but really there is only a very

minor LDB1 peak centred at that elution time. LDB1 and LMO2 are predominantly eluting in a complex centred at 12-12.5 ml (that must contain proteins that are not being blotted for here). Overall, this figure doesn't really seem to support the authors' thesis.

A2.1 Reviewer 2 is in agreement with Reviewer 1 in that they both suggest that Figure 2B does not add weight to support the thesis. As indicated above in answer to 1.1 we have therefore moved Fig. 2B to Supplementary Fig.1 and we have modified the text to suggest that because of the resolution of the technique, the data does not support but is compatible with our thesis:

LINE 252 "Although the elution pattern of JARID1A overlaps in many fractions with that of GATA1 and SCL, bands can also be seen for LMO2 and LDB1 suggesting that the resolution of the technique does not allow an accurate delineation of the composition of such JARID1A containing complexes."

Q2.2 Next, the authors show coIP experiments, IP-ing each of the relevant proteins in turn. Interactions between JARID1A and GATA1 and SCL are indicated (though they may be direct or indirect at this stage), but no significant interaction with LDB1 or LMO2 is seen (though the authors take the view that LMO2 *is* interactions – seems pretty marginal though).

A2.2 As suggested by Reviewer 2, the interaction with LMO2 is at the detection limit of the technique. We have therefore removed any statement that suggests interaction with LMO2.

Q2.3 In the last part of the manuscript, the authors use analytical ultracentrifugation to try to determine whether a domain from JARID1A (they choose the three PHD domains for their experiments) interacts directly with the structured domains from either GATA1 (the double ZF domain) or SCL (the bHLH domain), the two proteins that gave positive results in the coIPs (although it is odd that the authors claim that LMO2 is an interacting protein from the coIPs but then do not test this protein by AUC – specifically by sedimentation velocity).

A2.3 This is now addressed in **A2.2**.

Q2.4 First, the authors show AUC data for each of the three PHD domains with the GATA-NFCF and with SCL-bHLH [see below for comments about whether the SCL is a GST fusion protein...]. They see a clear interaction between PHD1 and SCL, but dismiss this as "weak transient or unspecific" because they say that they see three peaks at 1.8 (SCL), 3.9 (GST-PHD1) and 4.5 (complex) S. Now, there is a bit of ambiguity in Fig 4 because it seems that the panels somehow in the wrong place, but I can't see any panels that have a peak distribution like that for the mixture. The one that is labelled as PHD1 + SCL shows two peaks for the mixture – I can't see any free GST-1PHD1 peak in the panel (blue trace in panel D). As far as I can make out, this suggests that there was an excess of SCL in the mixture that was prepared, but that all of the PHD1 has been converted to a complex.

A2.4 We apologize for the lack of clarity. Figure 4 was mislabelled (see **A1.4**) and confusion was created by the fact that some of the data being referred to by Reviewer 2 were in fact to be found in the Supplementary section and not in Fig.4D. To address this we have now moved the data contained in supplementary Figures 1 and 2 into the main section and split the data into two Figures: Fig.3, containing all of the AUC analysis of SCL/E47 with PHD1,2 and 3; Fig.4, containing all of the AUC analysis of GATA1 with PHD1,2, and 3. We are confident that these new figures will aid the readership in the correct interpretation of the data.

Q2.5 And – even if there *were* three peaks, I don't think it is appropriate to dismiss the interaction as unspecific, just because the affinity might be weak – many biologically relevant interactions have affinities of even as low as mM (eg electron

transport proteins). In the area of gene regulation, many acetyllysine-bromodomain interactions are 10-100 μM . As an aside, these data highlight another issue with the approach taken here. There is no attempt to quantify the interactions (sed. velocity isn't really suited to affinity measurements, as far as I know) – so one can't even really say whether a situation that gave rise to three peaks would correspond to a 'weak' interaction because we don't get any sense of what 'weak' would mean in this context.

A2.5: We agree with Reviewer 2 and we have now re-phrased this comment in the text to eliminate references to unspecific or weak interactions:

LANE 296 "The presence of three peaks suggests a mixture of different states and a dynamic interaction between SCL/E47 and GST PHD1".

Q2.6 Continuing on in Figure 4, PHD2 clearly doesn't form a complex with SCL. The data with PHD3 are interpreted as not showing an interaction. The sed. coeff. is clearly a bit larger for one of the peaks in the mixture (in contrast to the PHD2 data, which are pretty clear), but I don't have a good enough sense of the uncertainties associated with sed. velocity data to know whether this difference is significant or not.

A2.6 We interpret the data as suggesting that there is no interaction between SCL/E47 and PHD3 based on the $c(s,ff0)$ analysis (which is now in Figure 3E). The $c(s,ff0)$ analysis provides a more accurate modelling of the sedimentation boundary as it more explicitly allows for the diffusion coefficients of the different species in the sample, which reflect their frictional coefficients and allow the computation of molecular weight values for individual species. For example, it does not assume single frictional coefficients for the any given species s value, but plots them in a third dimension as $c(s,ff0)$ which indicates the probability of each $c(s)$ species having a particular $f/f0$ i.e. diffusion coefficient i.e. weight. It is therefore a more sensitive form of analysis than $c(s)$ alone and would be expected to detect minor species better.

Q2.7 Finally, in Figure 5, the authors show sed. velocity data for a mixture of GATA1-NFCF and PHD2, which nicely show the formation of a complex. Overall, the concerns that I have raised above and below mean that I do not believe that this manuscript is suitable for publication as it stands.

As well as the various mixups that I mention,

A2.7 We hope we have addressed these mixups now as detailed in the various points above

Q2.8 my main concerns are (a) that the gel filtration data really don't support the authors' claims

A2.8 We hope we have now addressed this issue, as detailed in **A1.1** and **A2.1**.

Q2.9 and that (b) the authors do not show any data that speak to the correctly folded nature of the domains that they work with in the AUC experiments.

A2.9 The $c(s,ff0)$ values for all the proteins used in AUC are between 1 and 2. This along with the resolution of proteins in tight peaks (as can be seen in Fig. 3 and 4) suggests a globular nature of all the proteins used. Specifically, we believe that the three protein constructs SCL/E47BHLH, GATA1NFCF and JARID1A PHD2, which we claim can interact, are correctly folded because:

SCL/E47BHLH: The protein construct used in this study is the same that was crystallized and solved in El Omari et. 2013 Cell Rep. 4. 135-147 (**PDB ID 2YPB**).

GATA1NFCF: The protein construct used in this study is the same that was crystallized and solved in Wilkinson-White L. et al. 2015 Protein Sci.24 1649:1659. (**PDB ID:3VD6**)

JARID1A PHD2: To bring further evidence to the claim that this protein construct is folded we characterized JARID1A PHD2 by NMR spectroscopy. The spread of signals in the $1\text{H}-15\text{N}$ spectra, as can be seen in Supplementary Fig.2 suggests that the protein is well folded. We refer to this in the text in LINES 326-329.

Additional comments

Q2.10 Note that there is an inconsistency in the text regarding the state of the PHD domains. Figure 4 (and the Results section) indicates that the PHD domains are present as GST fusions in the AUC, whereas the M+M text describes them as being cloned into as His-tag vector. No M+M is given for the generation and purification of GST-fusion proteins.

A2.10 The text in the result sections is correct in that the PHD domains have been in cloned in both His-tag and GST-tag vectors. It is true however that we had omitted to include details on the cloning and purification of the GST-tag vectors in the Material and Methods Section. We have now rectified this.

Q2.11 If the PHD domains *are* GST fusions (and their sedimentation coefficients seem to indicate that this is the case), this raises some concerns in the AUC, given the micromolar dimerization constant

A2.11 Although all PHD domains were GST-tagged not all showed interactions with JARID1A. Furthermore, as indicated in the text, to exclude any interference of the GST affinity tag, we further probed the interaction between the PHD2 domain of JARID1A and GATA1NFCF.

Q2.12 I think we really need to see some sort of indication that the purified domains are correctly folded – such as a 1D 1H NMR spectrum, for example – or, in the case of GATA1 and SCL domains, an EMSA.

A2.12 As suggested by Reviewer 2, we have used 2D HSQC¹⁵N NMR to assess the correct folding of JARID1A PHD2 as detailed in **A2.9**. The correct folding of SCL/E47bHLH and GATA1NFCF had been shown previously in peer-reviewed publications as detailed in **A2.9**.

Minor points

Q2.13 Summary sentence 1 seems almost a tautology: ‘transcriptional regulation controls gene expression’

A2.13: Sentence 1 has been rephrased to avoid tautology into:” Chromatin remodelling and transcription factors play important roles in lineage commitment and development through control of gene expression.”

Q2.14 Line 70 – the authors need to fix “need Wadman et al. Embo 1997”

A2.14: This has been corrected.

Q2.15 M+M – spaces needed before units.

A2.15: This has been corrected.

Q2.16 M+NM – the authors need to describe how their nuclear extracts were prepared.

A2.16: This is described in the Materials and Methods section called “Nuclear extract preparations”.

Q2.17 The caption for Figure 1 needs to state whether the numbering shown is for the human proteins or another species.

A2.17: This has been corrected to include the species.

Q2.18 Line 470 – correct “Western blotting Western blotting analysis” (add full stop)

A2.18: This has been corrected.

Q2.19 Figure 3 – the authors should show sufficient of the western blots to include the nearest molecular weight marker – and indicate this marker.

A2.19: This has been corrected.

Q2.20 Line 243 – what do the authors mean by “post fusion”?

A2.20: The authors meant “following fusion”. This has been corrected.

Q2.21 Line 249 – the authors need to add a reference.

A2.21: A reference to Wang et al., 2010 has now been added.

Q2.22 The panels of Figure 4 are incorrectly arranged, compared to the figure legend and the article text.

A2.22: This has now been addressed as described in **A1.3** and **A2.4**

Q2.23 The authors do not indicate the molar concentrations of the proteins used in AUC shown in Figure 4 (only that shown in Fig 5). Were they the same in every experiment in Figure 4 and 5? They state only in the M+M that two of the proteins were used at 1 mg/ml (which isn't really the best units to use, given that it is the molar concentrations that are relevant).

A2.23: We apologize for the lack of clarity. All samples in the AUC were at the same molar concentration of 0.04mM. This has now been correctly detailed in the Materials and Methods Section.